# Semantic Arithmetic Coding Using Synonymous Mappings

**DOI:** 10.3390/e27040429

**Published:** 2025-04-15

**Authors:** Zijian Liang, Kai Niu, Jin Xu, Ping Zhang

**Affiliations:** 1Key Laboratory of Universal Wireless Communications, Ministry of Education, Beijing University of Posts and Telecommunications, Beijing 100876, China; liang1060279345@bupt.edu.cn (Z.L.); xujinbupt@bupt.edu.cn (J.X.); 2Department of Broadband Communication, Pengcheng Laboratory, Shenzhen 518055, China; 3State Key Laboratory of Networking and Switching Technology, Beijing University of Posts and Telecommunications, Beijing 100876, China; pzhang@bupt.edu.cn

**Keywords:** semantics, arithmetic coding, synonymity, synonymous mappings, semantic lossless compression

## Abstract

Recent semantic communication methods explore effective ways to expand the communication paradigm and improve the performance of communication systems. Nonetheless, a common problem with these methods is that the essence of semantics is not explicitly pointed out and directly utilized. A new epistemology suggests that synonymity, which is revealed as the fundamental feature of semantics, guides the establishment of semantic information theory from a novel viewpoint. Building on this theoretical basis, this paper proposes a semantic arithmetic coding (SAC) method for semantic lossless compression using intuitive synonymity. By constructing reasonable synonymous mappings and performing arithmetic coding procedures over synonymous sets, SAC can achieve higher compression efficiency for meaning-contained source sequences at the semantic level and approximate the semantic entropy limits. Experimental results on edge texture map compression show a significant improvement in coding efficiency using SAC without semantic losses compared to traditional arithmetic coding, demonstrating its effectiveness.

## 1. Introduction

In recent years, research on semantic communications has taken a different development route from traditional communication technologies. While traditional communications separately optimize source compression and data transmission guided by Shannon’s classical information theory [1,2], recent works on semantic communications mostly focus on exploring end-to-end performance optimization utilizing deep neural networks. As a typical method, deep joint source-channel coding (JSCC)-based methods effectively improve the end-to-end performance of communication systems for diverse source modalities, whether in point-to-point [3,4,5,6,7,8] or multi-user transmission scenarios [9,10,11,12,13]. However, these works are empirical in processing semantic information: they often refer to the high-dimensional space representation from neural network processing as “semantic information” [3,14], following expressions in the field of natural language processing (NLP) or computer vision (CV). Additionally, the optimization direction of these works still relies on measures in classical information theory, such as mean-squared error (MSE) [6], cross-entropy [4], and Kullback–Leibler (KL) divergence [7]. They do not deeply explore the concept of semantic information or guide information processing based on specific semantic information viewpoints and theoretical frameworks.

As a critical cornerstone of semantic communication, a complete semantic information theory framework can provide theoretical guidance for evaluating semantic abstraction abilities, measuring semantic processing bounds, and aiding in designing semantic communication methods [15]. In particular, it can help us recognize the problems that still exist in semantic information processing but have been overlooked due to the current trend in the evolution of semantic communication methods, such as how to identify the core of specific semantic information and how to approach its compression limits.

However, compared with the rapid development of semantic communication methods, semantic information theory still lacks a unified viewpoint, which makes the problems hard to solve. Since Weaver discussed three-level communication problems [16] in the 1950s, researchers have explored semantic information theory from various viewpoints, ranging from the perspective of logical probability [17,18,19,20] to fuzzy information theory [21,22,23]. Nevertheless, none of these theories has become a universal guiding theory for semantic communication methods, as these viewpoints face challenges in general scenarios. As theoretical research deepens, some basic viewpoints have gained consensus, including, but not limited to, the existence of background knowledge (referred to as the knowledge base) [20,24,25,26] and the hiddenness/invisibility of semantic information (referred to as the meaning behind the message) [27,28]. Based on these consensuses, various sub-viewpoints have been proposed to interpret semantic information in different directions. For example, some works suggest that the semantic information of the message can be extracted based on the knowledge base [29], and some works suggest that semantic information is closely tied to the selection of specific downstream tasks [14,30]. Therefore, various mathematical models have been established based on these viewpoints, and different forms of semantic entropy have thus been defined. As a result, there is still no unified definition of semantic entropy (unlike Shannon entropy), and most definitions lack the operational relevance that Shannon entropy has to many engineering problems [31], which makes it difficult to characterize the theoretical bounds of semantic information compression.

Based on the above situation, we recognize that a unified semantic information theory should be simple and highly abstract while guiding engineering exploration, especially in semantic information compression. With the proposal of a semantic information theory framework based on synonymity [32] (The official version was published in the *Journal on Communications*, Volume 6, 2024. Due to issues with the hyperlink provided by its DOI number, please refer to the following web link: https://www.joconline.com.cn/en/article/doi/10.11959/j.issn.1000-436x.2024111/, accessed on 25 June 2024), theoretical unification may become possible. Founded on commonly existing synonymous relationships in source data across diverse modalities and different task-oriented scenarios, this viewpoint establishes the framework based on the “synonymous relationship” between semantic information and syntactic information and constructs a complete semantic information theory system, including basic measures and coding theorems. The basic viewpoint of synonymity aligns with Shannon’s 1953 viewpoint [33], which defines the actual information of a stochastic process as “that which is common to all stochastic processes which may be obtained from the original by reversible encoding operations”. Additionally, the author believes previous viewpoints can be considered as different synonymous relationships within this theoretical framework and further proves that the proposed semantic information theory is compatible with classical information theory—specifically, that the synonymous set only contains one syntactic message. Such a viewpoint makes it possible to establish a unified semantic entropy measure, design a corresponding semantic entropy algorithm, and improve compression efficiency while maintaining specific semantic information. We observed that similar ideas have appeared in semantic compression methods for tabular data [34,35,36]; however, we should point out that this methodology should be adaptable to any source data type under the premise of well-designed synonymous mappings.

In this paper, we focus on designing a semantic entropy coding algorithm to improve compression efficiency while maintaining common semantic information determined by synonymous mappings and analyzing their theoretical limits. Specifically, we propose semantic arithmetic coding (SAC), an arithmetic coding (AC) method for semantic lossless compression built on synonymous mappings. The contributions of our work are as follows:**A semantic arithmetic coding algorithm built on synonymous mappings.** By constructing reasonable synonymous mappings to partition synonymous sets and determine the common semantic information, SAC performs the arithmetic encoding procedures over the synonymous set corresponding to the coding syntactic symbol, thereby achieving higher compression efficiency under intuitive semantic lossless conditions. Based on the corresponding decoding and post-processing processes, the decoded sequence preserves the common semantic information of the source data while reducing syntactic distortion. The computational complexity analyses of the SAC encoding and decoding procedures are also provided accordingly.**Semantic compression limit analysis of bidirectional squeezing.** The theoretical limit approachability to the semantic entropy of our proposed SAC is validated based on an extension of the code length theorem of arithmetic codes. This shows that, with an ideal semantic arithmetic code design, a compression efficiency very close to the semantic entropy limit built on synonymity can be achieved.**Performance verification on edge texture map datasets.** By designing appropriate synonymous mapping rules for the processing units on the BIPEDv2 edge texture feature map dataset [37] and using the proposed semantic arithmetic code for compression, we achieve a 20.5% improvement in average compression efficiency over classical arithmetic coding, which strictly requires syntactic losslessness. This is achieved while maintaining consistency with the semantic information of the original edge texture image and ensuring low syntactic distortion.

Herein, we emphasize that our work focuses on using the proposed semantic arithmetic code to approximate the corresponding semantic compression limit under reasonable synonymous mapping. How to construct a more effective synonymous mapping to minimize the semantic compression limit, as another key issue in semantic information theory [32], requires more future work.

The rest of this paper is organized as follows. Section 2 presents the system model of the semantic lossless source coding and introduces semantic entropy based on synonymous mappings. In Section 3, we present semantic arithmetic coding in detail, including its encoding procedure, decoding procedure, optional syntactic post-processing, and corresponding theoretical limit analysis on the average code length. Section 4 uses the edge texture semantics of natural images as the object, applies SAC for semantic lossless encoding and decoding, and verifies SAC’s effectiveness through numerical results. Finally, we conclude this paper in Section 5.

## 2. System Model and Theoretical Limits

In this section, we briefly review the system model of semantic lossless source coding, along with its theoretical compression limit based on a critical feature of synonymous mappings.

### 2.1. System Model

Semantic lossless source coding is an extension of classic lossless source coding under the guidance of semantic information theory, with its goal still being to compress source data. However, unlike classic lossless source coding, semantic lossless source coding focuses on ensuring no distinctions in implicit meanings between sequences before encoding and after decoding, without strictly requiring complete consistency in their explicit syntactical forms.

As stated in [32], all perceptible messages are syntactic information, and all such syntactic information is presented to illuminate the underlying semantic information. Therefore, we can establish the system model for semantic source coding as follows:(1)$$\tilde{u}\overset{f\left(\cdot \right)}{\to }u\overset{e\left(\cdot \right)}{\to }b\overset{d\left(\cdot \right)}{\to }\hat{u}\overset{g\left(\cdot \right)}{\to }\hat{\tilde{u}},$$
where $\tilde{u}$ and $\hat{\tilde{u}}$ are the invisible source and reconstructed semantic variable sequences, and u and $\hat{u}$ are the perceptible source and reconstructed syntactic variable sequences, respectively. We assume that *m* represents the length of the syntactic sequences u and $\hat{u}$. The mapping $f\left(\cdot \right)$ and its reverse $g\left(\cdot \right)$ represent the invisible conversion relationship between the semantic information and the syntactic information, i.e., the synonymous mappings and synonymous de-mappings presented by [32].

For the main process of the coding, the semantic source encoder $e\left(\cdot \right)$ operates on the syntactic sequence u, encoding it into a codeword sequence b of length *l*, and the corresponding semantic source decoder $d\left(\cdot \right)$ transforms the codeword sequence b into the reconstructed syntactic sequence $\hat{u}$. Only consistency between the semantic sequences $\tilde{u}$ and $\hat{\tilde{u}}$ needs to be guaranteed in the coding procedures; thus, the lossless constraints between the syntactic sequences u and $\hat{u}$ can be relaxed, which makes the coding a lossy source coding from a syntactic perspective.

Herein, we state that, in general cases, synonymous mappings (constructed from a single piece of semantic information to multiple syntactic representations) must first be applied to the sequence data to build non-overlapping synonymous sets, such as natural languages, images, and audio sequences. This is because when the information sequence is sufficiently long, contextual structures and characteristics such as Markov chains [38] can help determine the unique meaning of polysemous or homonymous words within the sequence. To illustrate this statement, we provide an example using the following English sentence as the data source u:

“I walk along a river bank”.

In this sentence, most words have a single meaning, while the word “bank” itself has multiple meanings, such as a financial institution and the side of a river. However, when placed in this sentence (especially after the word “river”), the meaning of the word “bank” collapses into a specific interpretation, i.e., “the side of the water”.

Based on this simple example, we can summarize a basic principle for determining synonymous mapping relationships: **It should first be considered at the sequence level and then applied to individual elements based on contextual relationships within the sequence for practical implementation**. This principle means the following:If a sequence cannot determine a unique meaning for each element, the sequence length must be extended until its context is sufficient to make this determination, and then treating the extended sequence as the processing object;Conversely, if a definite meaning can be independently determined for each element in the sequence, the synonymous mappings can be directly applied to each element;Furthermore, if all elements in the sequence follow an i.i.d assumption, in which each element represents diverse meanings under the same attribute, then each element in the sequence can be assigned the same synonymous mapping.

We emphasize that the principle of first considering synonymous mappings at the sequence level cannot only be derived from simple examples but can also be rigorously proven based on the semantic asymptotic equipartition property (AEP) proposed in [32] under the assumption of sufficiently long syntactic sequences, providing sufficient theoretical support.

In view of this, the mathematical model and algorithm design for establishing synonymous relationships for each variable in the sequence, as described in the following text, adhere to this principle.

### 2.2. Synonymous Mappings-Based Semantic Entropy

As remarked in [32], synonymity is a critical source of the relationships between semantic information and syntactic information since, in most instances, single-meaning semantic information has myriad syntactic presentation forms. However, it should be emphasized that directly processing semantic information is not feasible, as semantic information is hidden behind the message (syntactic information) and thus invisible. Thus, semantic information processing must construct entities that can reflect semantic content. Based on the synonymous relationship we previously established, syntactic information with similar meanings can be grouped into synonymous sets, which serve as entities that reflect semantic information. Therefore, processing information at the level of these entities is then equivalent to processing semantic information. In light of this, for semantic information processing, constructing synonymous sets based on synonymous relationships is essential.

Figure 1 shows an example of the synonymous mappings $f_{i}:{\tilde{U}}_{i}\to U_{i}$ between the semantic information set ${\tilde{U}}_{i}$ and the syntactic information set $U_{i}$ for the *i*-th variable in the source sequences $\tilde{u}$ and u. From this example, a general rule can be observed: semantic elements can be mapped to an equal number of synonymous sets that represent different meanings and contain all the possible syntactic values without overlap between any two synonymous sets.

For an i.i.d semantic sequence $\tilde{u}$ with unified $f:\tilde{U}\to U$ for ∀i=1,2,…,m, the semantic entropy $H_{s}\left(\tilde{U}\right)$ is expressed as(2)$$H_{s}\left(\tilde{U}\right)=-\sum_{k}p\left(U_{k}\right)\log p\left(U_{k}\right),$$
where the probability of the *k*-th synonymous set is(3)$$p\left(U_{k}\right)=\sum_{n\in N_{k}}p\left(u_{n}\right),$$
where $N_{k}$ denotes a set that contains the indices of the syntactic values with the same meaning as the semantic element *k*. It should be noted that a synonymous relationship itself is a form of a semantic knowledge base, which is consistent with the current consensus: by applying specific prior knowledge or criteria to determine synonymous relationships for actual source information, all syntactic data within the constructed synonymous sets share common information corresponding to that prior knowledge or criteria.

In [32], we demonstrated that for single-symbol semantic source coding, with the semantic prefix code performed over the synonymous sets, the average code length can approach the theoretical semantic entropy limit $H_{s}\left(\tilde{U}\right)$ without semantic losses by providing a theorem based on the semantic Kraft inequality. Naturally, the same effect can be achieved by performing semantic prefix coding on the sequences, leading to our proposed semantic arithmetic coding.

## 3. Semantic Arithmetic Coding

Consider a sequence compression procedure with arithmetic coding methods for the syntactic sequence u, in which each syntactic variable $u_{i}$ exhibits a similar synonymous relationship to that shown in Figure 1, based on the basic determining principle. Traditional arithmetic coding directly performs the coding procedure on each syntactic variable $u_{i}$ without considering the implicit meaning, thereby lacking certain compression efficiency due to the sole requirement of semantic losslessness. In this section, we propose semantic arithmetic coding (SAC) using synonymous mappings for efficient semantic compression with intuitive semantic synonymity.

### 3.1. The Encoding Procedure

Figure 2 shows the general framework of the SAC encoding procedure. Similar to the traditional method [2,39], the SAC encoder uniquely maps the message u to a sub-interval on the $\left[0,1\right)$ interval and outputs the shortest codeword represented by a binary fraction *b* in this sub-interval as the encoding result. The difference is that, to achieve semantic-oriented compression, the SAC encoder constructs reasonable synonymous mappings to determine the synonymous set $U_{i,r_{i}}$ for each syntactic variable $u_{i}$ and performs the coding interval update procedure over each determined synonymous set $U_{i,r_{i}}$.

As a specific description of the process shown in Figure 2, the encoding process of the SAC encoder is detailed in Algorithm 1. With an initialized encoding interval $\left[L_{0},H_{0}\right)=\left[0,1\right)$ and an interval length of $R_{0}=1$, the SAC encoder performs semantic compression through a series of iterations, in which each iteration *i* includes the following:*Construct synonymous mappings:* For all the syntactic values $\left\{u_{i,n}\right\}$ of the *i*-th variable, construct synonymous mappings $f_{i}:{\tilde{U}}_{i}\to U_{i}$ under specific semantic criteria to partition synonymous sets ${\left\{U_{i,k}\right\}}_{k=1,\ldots ,|{\tilde{U}}_{i}|}$, in which $U_{i,k}={\left\{u_{i,n}\right\}}_{n\in N_{k}}$;*Determine synonymous sets:* According to the actual value of the syntactic variable $u_{i}$, determine a synonymous set $U_{i,r_{i}}$ such that $u_{i}\in U_{i,r_{i}},\;r_{i}\in \left\{1,\ldots ,|{\tilde{U}}_{i}|\right\}$;*Calculate probabilities:* For all synonymous sets $\left\{U_{i,k}\right\}$, calculate their probabilities with(4)$$p\left(U_{i,k}\right)=\sum_{n\in N_{k}}p\left(u_{i,k}\right).$$*Update encoding interval:* According to the determined synonymous set $U_{i,r_{i}}$ and the probabilities for all synonymous sets $p\left(U_{i,k}\right),k=1,\ldots ,|{\tilde{U}}_{i}|$, update the encoding interval $\left[L_{i},H_{i}\right)$ and its interval length $R_{i}$ with(5)$$\left\{\begin{array}{cc} L_{i}=L_{i-1}+\sum_{k=1}^{r_{i}-1}p\left(U_{i,k}\right), & \\ H_{i}=L_{i}+p\left(U_{i,r_{i}}\right)\cdot R_{i-1}, & \\ R_{i}=p\left(U_{i,r_{i}}\right)\cdot R_{i-1}=H_{i}-L_{i}. \end{array}\right.$$

Once the encoding interval update process corresponding to the last variable $u_{m}$ is completed, the SAC encoder concludes its iterations. Then, it determines a shortest binary fraction $b=\left(b_{1},\ldots ,b_{l}\right)$ as the output codeword such that its corresponding decimal fraction *c* belongs to the final interval $\left[L_{m},H_{m}\right)$ and satisfies(6)$$c=b_{1}\cdot 2^{-1}+b_{2}\cdot 2^{-2}+\ldots +b_{l}\cdot 2^{-l}.$$

Finally, the SAC encoder transmits the output codeword b to the receiver for reconstructing the syntactic sequence. As the necessary information, the length *m* of the syntactic sequence u, along with the synonymous set partitions of each syntactic variable and their corresponding probability information, must be synchronized at the receiving end.
**Algorithm 1:** Encoding algorithm of SAC encoder
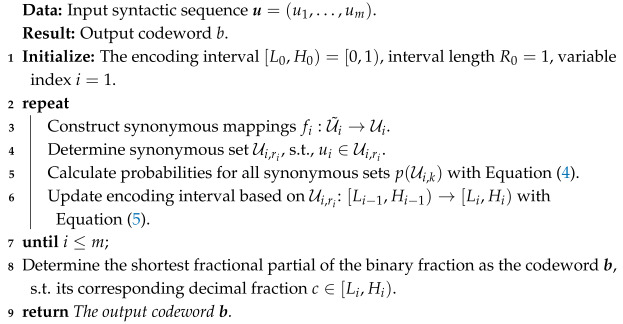


When only the i.i.d. assumption is considered, the possible values of each syntactic variable, the synonymous mappings along with the synonymous set partitions, and the corresponding probabilities are exactly the same. In this case, the process of SAC encoding is equivalent to a simplified procedure that first converts the syntactic sequence into a synonymous set sequence using the predefined synonymous mapping rules, then treats the synonymous subset sequence as a new syntactic sequence and compresses it using traditional arithmetic coding.

Herein, we provide a complexity analysis of the SAC encoding process and compare it with the general arithmetic encoding process.

First, we consider the standard case of SAC encoding. Considering that the number of syntactic values for each variable may be different, as well as the number of synonymous sets, we assume that the average number of syntactic values per variable in the sequence is $|\bar{U}|$, and the average number of corresponding synonymous sets is $|\bar{\tilde{U}}|$. These two averages are used to evaluate the encoding complexity.

For a syntactic source sequence with length *m*, the computational complexity of calculating the probabilities of the synonymous sets for each variable is expressed as $O(|\bar{U}|\cdot m)$. Additionally, the complexity of updating the coding interval is expressed as $O(|\bar{\tilde{U}}|\cdot m)$ since only the scale of the synonymous sets is considered. Therefore, the computational complexity of SAC can be summarized as $O((|\bar{U}\left|+\right|\bar{\tilde{U}}\left|)\cdot m\right)$, which is slightly higher than the complexity of general arithmetic encoding $O(|\bar{U}|\cdot m)$.

However, for a special case when only the i.i.d. assumption is considered, the number of syntactic values is |U|, and the number of corresponding synonymous sets is $|\tilde{U}|$ for all the variables in the sequence. In this case, the synonymous mappings can be calculated once, regardless of the sequence length *m*. So, the computational complexity of SAC encoding is reduced to $O(|U|+|\tilde{U}\left|\cdot m\right)\approx O(|\tilde{U}\left|\cdot m\right)$, which is less than the complexity of general arithmetic encoding O(|U|·m) since $|\tilde{U}\left|\leq |U\right|$.

### 3.2. The Decoding Procedure

As a duality process within SAC encoding, the decoding procedure of the SAC decoder is presented in Algorithm 2. To reconstruct the syntactic sequence from the received codeword b, the SAC decoder initializes a decoding interval $\left[L_{0}^{\prime },H_{0}^{\prime }\right)=\left[0,1\right)$ with an interval length of $R_{0}^{\prime }=1$ and then utilizes the decimal fraction *c* corresponding to the codeword b to determine the values of each syntactic variable. The determination process is also performed through a series of iterations, in which each iteration *i* includes the following:*Construct synonymous mappings:* For all the syntactic values $\left\{{\hat{u}}_{i,n}\right\}$ of the *i*-th reconstructed variable, construct opposite synonymous mappings $g_{i}:{\hat{U}}_{i}\to {\hat{\tilde{U}}}_{i}$ according to the synchronized synonymous sets information at the sending end, and accordingly partition synonymous sets ${\left\{{\hat{U}}_{i,k}\right\}}_{k=1,\ldots ,|{\tilde{U}}_{i}|}$, in which ${\hat{U}}_{i,k}={\left\{{\hat{u}}_{i,n}\right\}}_{n\in {\hat{N}}_{k}}$;*Synchronize probabilities:* To guarantee successful semantic decoding, the probabilities of all synonymous sets should be synchronized with the sending end, i.e., $p\left({\hat{U}}_{i,k}\right)=p\left(U_{i,k}\right)$. The probability of each syntactic value needs to be assigned, satisfying(7)$$p\left({\hat{U}}_{i,k}\right)=\sum_{n\in {\hat{N}}_{k}}p\left({\hat{u}}_{i,k}\right).$$*Determine synonymous sets:* Based on the decimal fraction *c* and the decoding interval $\left[L_{i-1}^{\prime },H_{i-1}^{\prime }\right)$, determine the reconstructed synonymous set ${\hat{U}}_{i,r_{i}},\;r_{i}\in \left\{1,\ldots ,|{\tilde{U}}_{i}|\right\}$ such that(8)$$\frac{c-L_{i-1}^{\prime }}{R_{i-1}^{\prime }}\in \left[\sum_{k=1}^{r_{i}-1}p\left({\hat{U}}_{i,k}\right),\sum_{k=1}^{r_{i}}p\left({\hat{U}}_{i,k}\right)\right).$$*Export syntactic value:* Select a syntactic value ${\hat{u}}_{i,n}$ from the determined synonymous set ${\hat{U}}_{i,r_{i}}$ as the reconstructed syntactic value ${\hat{u}}_{i}$. It can be selected based on certain rules, such as the maximum probability-based selection within the synonymous set, normalized probability-based random selection based on the normalized probability of each syntactic value in the determined synonymous set(9)$$p^{\prime }\left({\hat{u}}_{i,n}\right)=\frac{p\left({\hat{u}}_{i,n}\right)}{p\left({\hat{U}}_{i,r_{i}}\right)},n\in N_{r_{i}},$$
or context correlation-based selection based on already-reconstructed syntactic sequences.*Update decoding interval:* Based on the determined synonymous set ${\hat{U}}_{i,r_{i}}$ and the probabilities for all synonymous sets $p\left({\hat{U}}_{i,k}\right),k=1,\ldots ,|{\tilde{U}}_{i}|$, update the decoding interval $\left[L_{i}^{\prime },H_{i}^{\prime }\right)$ and its interval length $R_{i}^{\prime }$ with(10)$$\left\{\begin{array}{cc} L_{i}^{\prime }=L_{i-1}^{\prime }+\sum_{k=1}^{r_{i}-1}p\left({\hat{U}}_{i,k}\right), & \\ H_{i}^{\prime }=L_{i}^{\prime }+p\left({\hat{U}}_{i,r_{i}}\right)\cdot R_{i-1}^{\prime }, & \\ R_{i}^{\prime }=p\left({\hat{U}}_{i,r_{i}}\right)\cdot R_{i-1}^{\prime }=H_{i}^{\prime }-L_{i}^{\prime }. \end{array}\right.$$


Once the decoding interval update process corresponding to the last reconstructed variable ${\hat{u}}_{m}$ is completed, the SAC decoder concludes its iterations and outputs a combination of reconstructed syntactic values $\hat{u}=\left({\hat{u}}_{1},\ldots ,{\hat{u}}_{m}\right)$ as the reconstructed syntactic sequence.

Similar to the SAC encoder when only the i.i.d. assumption was considered, the synonymous mappings, along with the synonymous set partitions, and the corresponding probabilities are exactly the same. In this case, the process of SAC decoding is equivalent to a simplified procedure that first reconstructs the synonymous set sequence with the traditional arithmetic decoder and determines each syntactic variable based on the unified synonymous set partition rules.

To summarize, SAC implements semantic compression and reconstruction by constructing reasonable synonymous mappings and performing arithmetic coding procedures over synonymous sets.

Herein, we provide a corresponding complexity analysis of the SAC decoding process and compare it with the general arithmetic decoding process.

**Algorithm 2:** Decoding algorithm of SAC decoder

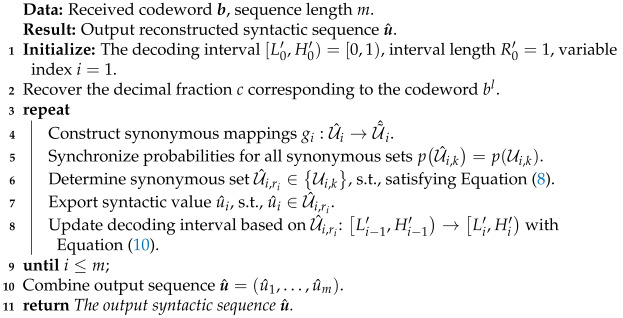



Similar to the encoding complexity analysis, we first consider the standard case of SAC decoding. we assume that the average number of syntactic values per variable in the sequence is $|\bar{U}|$, and the average number of corresponding synonymous sets is $|\bar{\tilde{U}}|$. These two averages are used to evaluate the decoding complexity. However, unlike the encoding side, since the probability and grouping information of the synonymous set can be directly transmitted from the encoding end to the decoding end, the decoding end does not incur additional computational complexity related to the probability calculation of the synonymous sets. Therefore, only the necessary computational complexity for the SAC decoding process should be considered.

For an expected decoded syntactic source sequence with the length *m*, the computational complexity of updating the coding interval is $O(|\bar{\tilde{U}}|\cdot m)$. Additionally, the complexity of the normalized probability calculation of each syntactic value should also be considered, which is expressed as $O((|\bar{U}\left|/\right|\bar{\tilde{U}}\left|)\cdot m\right)$, in which the division term indicates that only the syntactic values in the determined synonymous set should be used for the calculation of the conditional probability. Therefore, the computational complexity of SAC can be summarized as $O((|\bar{\tilde{U}}\left|+(\right|\bar{U}\left|/\right|\bar{\tilde{U}}\left|))\cdot m\right)$, which is slightly lower than the complexity of the general arithmetic encoding $O(|\bar{U}|\cdot m)$ since not all probability intervals of syntactic elements need to be considered.

As for the special case with i.i.d. assumption, the calculation of the normalized probability can be done only once as a reference for the overall sequence decoding, thus the complexity is expressed as O(|U|) (rather than the complexity with the division term, since every value should have its normalized probability calculated within its corresponding synonymous sets), regardless of the sequence length *m*. So, the computational complexity of SAC decoding can be further reduced to $O(|U|+|\tilde{U}\left|\cdot m\right)\approx O(|\tilde{U}\left|\cdot m\right)$, which is less than the complexity of general arithmetic encoding O(|U|·m) since $|\tilde{U}\left|\leq |U\right|$.

Therefore, we conclude that the computational complexity of SAC encoding is generally higher than that of general arithmetic encoding but lower than that in the i.i.d. case, while the computational complexity of SAC decoding is lower than that of general arithmetic decoding in all cases.

### 3.3. Optional Syntactic Post-Processing

While SAC can achieve semantic-level lossless encoding and decoding based on synonymous mapping, it may not ensure syntactic-level consistency of the reconstructed syntactic sequence with human perception since the syntactic-level accuracy requirement is relaxed. This is mainly because the aforementioned selection rules for reconstructed syntactic values cannot effectively utilize the overall correlation information within the syntactic sequence. Therefore, an effective solution is to utilize syntactic post-processing methods to optimize syntactic-level consistency by utilizing the overall contextual correlation information. The final output of syntactic sequences after post-processing is represented as(11)$$\tilde{u}=\min_{f_{p}\left(\hat{u}\right)}d\left(u,f_{p}\left(\hat{u}\right)\right)\;s.t.\;\forall i,\left\{{\tilde{u}}_{i},{\hat{u}}_{i}\right\}\in {\hat{U}}_{i,r_{i}},$$
where $f_{p}\left(\cdot \right)$ denotes a post-processing function and $d\left(\cdot ,\cdot \right)$ is a specific distortion measure. ${\tilde{u}}_{i}$ is the *i*-th element in the output sequence $\tilde{u}=\left({\tilde{u}}_{1},\ldots ,{\tilde{u}}_{m}\right)$, which should share the same synonymous set ${\hat{U}}_{i,r_{i}}$ with its input ${\hat{u}}_{i}$.

Specifically, the post-processing function $f_{p}\left(\cdot \right)$ can be determined through two types of operations: empirical linear processing and nonlinear processing based on deep neural networks. Empirical linear processing mainly relies on various linear smoothing filters in traditional signal processing to smooth parts of the reconstructed syntactic sequence that do not conform to perception. In contrast, nonlinear processing is mainly based on deep neural networks and various optimal algorithms. By constructing a dataset including the reconstructed syntactic sequences output by the SAC decoder as the input data and their corresponding source syntactic sequences as the labels and training the model to minimize the gap between the post-processing output sequence and the source syntactic sequence, the syntactic post-processing model can be applied after the SAC decoder to reduce syntactic-level distortion. With a hard decision module using suitable judgment criteria, the post-processed sequence can be finally determined.

It should be noted that not all semantic information processing scenarios require syntactic-level consistency. Therefore, syntactic post-processing is optional for SAC and only necessary to meet specific syntactic consistency requirements.

### 3.4. Theoretical Limit Analysis

Herein, we analyze the theoretical semantic compression limits of our proposed SAC, based on the extension of the code length theorem of the classical arithmetic coding algorithm [2] to the semantic version described below.

**Theorem** **1.** 
*For a semantic arithmetic coding procedure, given any syntactic sequence u of length m with the probability mass function of its corresponding synonymous set sequence $q\left(U_{1,r_{1}},\ldots ,U_{m,r_{m}}\right)$, it enables one to encode u in a code of length $-\log q\left(U_{1,r_{1}},\ldots ,U_{m,r_{m}}\right)+2$ bits.*


This theorem can be simply proved by replacing the probability mass function for the syntactic sequence in the compression limit of arithmetic coding [2] with the probability mass function for the synonymous set sequence, in which the compression limit of arithmetic coding is fundamentally derived from the corresponding analysis of Shannon–Fano–Elias codes [2,40].

Under an i.i.d. assumption and the assumed distribution *q* being equal to the true distribution *p*, the average code length can approach the semantic entropy limit $H_{s}\left(\tilde{U}\right)$ if m→∞, i.e.,(12)$$H_{s}\left(\tilde{U}\right)<{\bar{L}}_{s}\leq \lim_{m\to \infty }\frac{-\log p\left(U_{1,r_{1}},\ldots ,U_{m,r_{m}}\right)+2}{m},$$
where the upper bound approaches $H_{s}\left(\tilde{U}\right)$, demonstrating the theoretical compression limit of our proposed SAC can be tightly bounded around the semantic entropy, thereby verifying the achievability of the proposed method.

## 4. Implementations Showcase

In this section, we verify the semantic compression performance of our proposed SAC along with its ability to preserve intuitive semantic synonymity.

### 4.1. Experimental Setup

We consider a scenario of edge texture-oriented semantic compression for natural images. In our experiments, the edge texture maps and their corresponding natural images are all derived from the BIPEDv2 dataset [37]. We use the edge texture in natural images as a form of semantic information contained in the images and employ the annotated edge texture maps as the syntactic source for semantic compression and reconstruction. The resolutions of the annotated edge texture maps are 1280×720, in which pixels representing objects and the background are labeled as value 0, and those of the edge texture between different objects and between an object and the background are labeled as value 1.

To construct the synonymous mappings, we regard each non-overlapping 2×2-pixel block as a syntactic symbol so that the edge texture maps can be considered a syntactic sequence of length m=640×360=230,400. On this basis, we partition the 16 syntactic symbols into seven synonymous subsets based on Figure 3, in which each synonymous set represents a type of local edge texture semantic. For example, the first synonymous set $U_{i,1}$ (“almost empty texture”) includes all the all-zero and single blocks, while the last synonymous set $U_{i,7}$ (“upper-left (lower-right) direction texture”) includes all the blocks that can be regarded as being in the upper-left or lower-right. We assume that syntactic sequences satisfy the i.i.d. assumption and the probabilities of syntactic symbols. The synonymous sets in the encoding and decoding process are directly obtained based on probability statistics and are synchronized at both ends.

Figure 4 shows the overall SAC processing flow for edge texture maps of natural images. First, the SAC encoder compresses the source edge texture map u at the semantic level instructed by synonymous mappings and produces the output bitstream b. Then, the SAC decoder reconstructs the source information ${\hat{u}}^{\left(r\right)}$ using the bit stream b under the guidance of synonymous de-mapping. Finally, a neural network model is used for syntactic post-processing, followed by a hard decision module and an edge texture recovery module to obtain the final post-processed edge texture map ${\hat{u}}^{\left(p\right)}$.

During the SAC decoding process, the selection rules for each reconstructed syntactic value ${\hat{u}}_{i}^{\left(r\right)}$ are as follows:If the decoded synonymous set is $U_{i,1}$, the maximum probability value of 0 is output;If another synonymous set is decoded, the reconstructed syntactic value is randomly output based on the normalized probability within the synonymous set.

Regarding the post-processing model, a convolutional layer with a 2×2 kernel and a stride of 2 is first utilized for pixel block embedding, mapping each 2×2 pixel block to a latent vector with c=128 channels. Then, the model employs five ResNet blocks [41] to capture the overall spatial correlations of the texture map. The final projection layer maps each c×1×1 vector to a 16×1×1 vector and uses the softmax function to estimate the probabilities of each syntactic value shown in Figure 3 for $u_{i}$, thereby obtaining a probability-estimation sequence ${\tilde{u}}^{\left(p\right)}$ for the syntactic source u.

To reduce syntactic errors in the post-processed edge texture map under the given synonymous sets, we conduct a two-stage training process. The first stage optimizes the standard cross-entropy (CE) loss between the projection layer output ${\tilde{u}}^{\left(p\right)}$ and the original syntactic source u to capture its spatial correlation, i.e.,(13)$$L_{1}=\mathrm{CE}\left(u,{\tilde{u}}^{\left(p\right)}\right),$$
while the second stage further adjusts the standard cross-entropy loss to a synonymous cross-entropy (S-CE) loss with a constraint on the decoded synonymous set sequence, i.e.,(14)$$L_{2}=S\text{-}\mathrm{CE}\left(u,{\tilde{u}}^{\left(p\right)}\right).$$

The calculations of the CE and S-CE losses are almost identical, except for the probability estimation methods for each syntactic value. For the standard CE loss, the probability estimation for the syntactic value ${\tilde{u}}_{i}^{\left(p\right)}$ equals $n\in \left\{0,1,\ldots ,15\right\}$ (denoted as ${\tilde{u}}_{i,n}^{\left(p\right)}$), and it is presented using the softmax function, i.e.,(15)$$\mathrm{Softmax}\left({\tilde{u}}_{i,n}^{\left(p\right)}\right)=\frac{\exp\left({\tilde{u}}_{i,n}^{\left(p\right)}\right)}{\sum_{n=0}^{15}\exp\left({\tilde{u}}_{i,n}^{\left(p\right)}\right)},$$
while the probability estimation, under the constraints of a given synonymous set, is presented using an adjusted synonymous softmax (s-softmax) function, i.e.,(16)$$S\text{-}\mathrm{Softmax}\left({\tilde{u}}_{i,n}^{\left(p\right)}\right)=\frac{\exp\left({\tilde{u}}_{i,n}^{\left(p\right)}\right)}{\sum_{n\in {\hat{U}}_{i,r_{i}}}\exp\left({\tilde{u}}_{i,n}^{\left(p\right)}\right)}.$$

This S-SE function, along with its s-softmax function, follows the requirement of Equation (Equation 11), where the output syntactic value belongs to the decoded synonymous set, thereby achieving better syntactic-level consistency while maintaining semantic losslessness.

After training, by determining each value with the highest probability as the value of the syntactic symbol ${\hat{u}}_{i}^{\left(p\right)}$, we obtain the final post-processed edge texture map ${\hat{u}}^{\left(p\right)}$.

We use the Adam optimizer to train the post-processing model. During training, we first use the loss function $L_{1}$ for the first stage, with 100 epochs and a learning rate of $\mathrm{lr}=1\times 10^{-5}$, to estimate the probability in Equation (Equation 15), then modify the loss function to $L_{2}$ for the second stage, with 200 epochs and a learning rate of $\mathrm{lr}=5\times 10^{-6}$ to achieve a better estimation of the probability in Equation (Equation 16) under the constraints. After training, the average S-CE of each synset symbol on the test set is 0.014, which supports acceptable post-processed syntactic qualities.

### 4.2. Visualization Results and Numerical Analysis

We employed our proposed SAC method on the test set, which consisted of 50 edge texture maps, along with their corresponding natural images, to verify the semantic compression effect, and utilized the traditional AC method as the comparison scheme. As a representative result, Figure 5 shows an example of the compression and reconstruction effects for semantic compression of edge texture maps using our proposed SAC method, in which the source, reconstructed, and post-processed edge texture maps are all labeled on the corresponding natural image to verify the semantic accuracy of the edge texture semantics. From the perspective of the reconstruction effect, although the syntactic form of the reconstructed and post-processed edge texture maps differed from the original edge texture, they did not affect the accuracy of their edge texture semantics in this natural image. This observation is consistent with the effects seen in the other samples in the test set, indicating that no semantic losses exist on the reconstructed edge texture map with our proposed method.

Additionally, by comparing the reconstructed and post-processed edge feature maps, we found that the syntactic consistency of the reconstructed map with the source one was relatively poor. However, with the syntactic post-processing model, the syntactic distortion between the post-processed map and the source map was effectively reduced, translating to better consistency at the syntactic level. This confirms the effectiveness of our proposed syntactic post-processing model.

Based on these results, it is evident that compression efficiency was significantly improved using SAC. From the perspective of the actual code length, SAC provided effective compression efficiency improvements compared with the traditional method. For the example edge texture map shown in Figure 5, SAC saved 26,515 sebits [32] compared with traditional arithmetic coding, equivalent to a 12.9% improvement in compression efficiency. As for the entire test set, SAC saved an average of 29,085.1 sebits of code length compared to traditional methods, equivalent to a 20.5% improvement in the average compression efficiency. This demonstrates that SAC significantly improves compression efficiency compared to traditional arithmetic coding with semantic losslessness.

From another perspective, the averaged code length calculated by SAC can exceed the Shannon entropy of the classical information theory, and further approximate the theoretical semantic compression limits, i.e., semantic entropy. Specifically, for both the single example test edge texture map and the entire test set, the gap between the average code sequence length after SAC encoding and the corresponding ideal code length was within 2 bits, which confirms the analysis results in Section 3.4.

These results effectively demonstrate the performance of our proposed SAC method, i.e., it can achieve an effective compression efficiency improvement and approximate the semantic entropy with intuitive semantic losslessness.

## 5. Conclusions

In this paper, we propose a semantic source coding method called semantic arithmetic coding. By constructing reasonable synonymous mappings and performing arithmetic coding procedures over synonymous sets, the compression efficiency can be significantly improved compared with traditional arithmetic codes under intuitive semantic lossless conditions. Additionally, we provide a theoretical limit analysis of our proposed method based on an extension of the code length theorem of arithmetic codes, along with experimental verification, to confirm its convergence to semantic entropy. Future work will focus on theories and methods for constructing more effective synonymous mappings while ensuring intuitive semantic losslessness, thereby minimizing the semantic compression limit. In addition, integrating semantic arithmetic coding algorithms with neural network compression coding and extending the idea to the design of corresponding semantic communication methods are important directions for future work.

## 6. Patents

The work presented in this paper is related to a Chinese Patent [42] and a corresponding US patent [43].

## Figures and Tables

**Figure 1 entropy-27-00429-f001:**
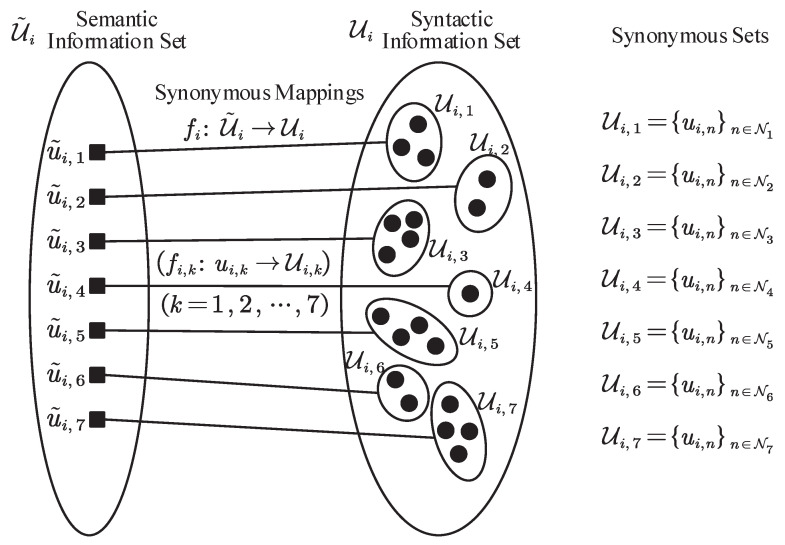
An example of the synonymous mappings and the corresponding synonymous sets.

**Figure 2 entropy-27-00429-f002:**
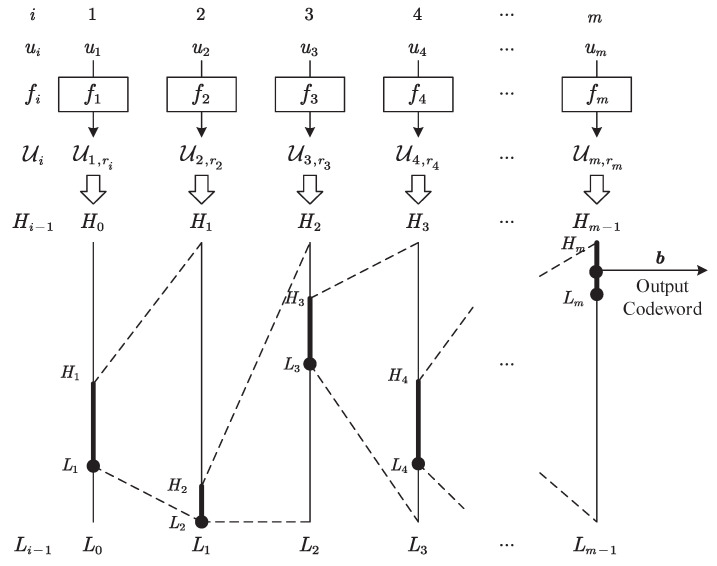
A schematic diagram of the SAC encoding procedure.

**Figure 3 entropy-27-00429-f003:**
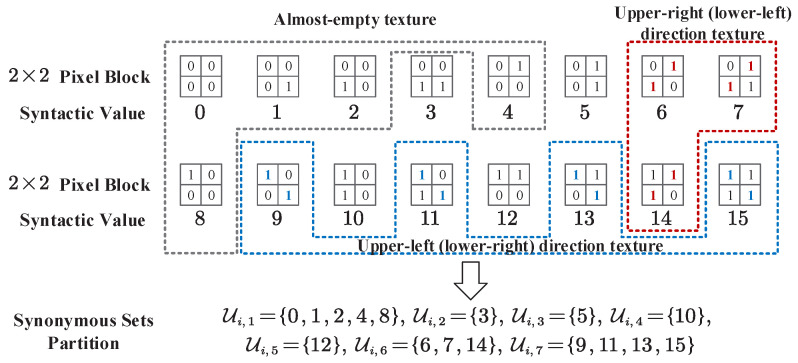
The synonymous set partitions for edge texture maps.

**Figure 4 entropy-27-00429-f004:**
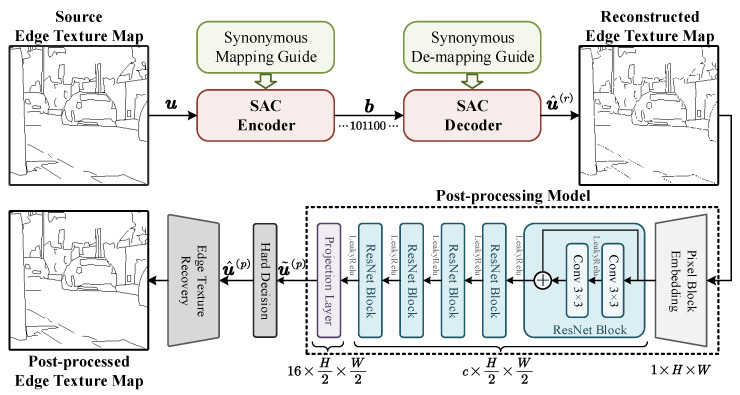
The overall SAC processing flow for edge texture maps of natural images.

**Figure 5 entropy-27-00429-f005:**
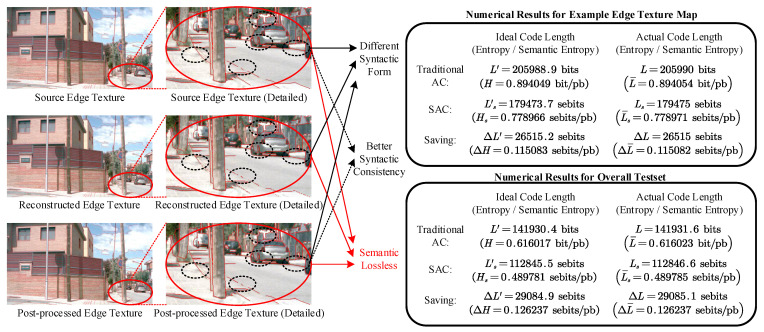
An example of compression and reconstruction effects for edge texture map semantic compression, in which “sebits” denotes the semantic bits for the resulting unit of semantic source coding, as presented in [32]. Additionally, “bit/pb” and “sebit/pb”, respectively, denote the bit per pixel block and the sebit per pixel block, acting as the units of entropy and the average code length of traditional AC and our proposed SAC based on our coding configuration.

## Data Availability

The data that support the findings of this study are available from the corresponding author upon reasonable request.

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
