# Peer review of "Semantic Arithmetic Coding Using Synonymous Mappings"

_entropy, 2025, doi:10.3390/e27040429_

Round 1
Reviewer 1 Report
Comments and Suggestions for Authors
Unfortunately, the submitted paper is poorly written. It also lacks convincing evidence for synonymity as a fundamental feature of semantic information (contrary to the authors' claims), as it does not sufficiently incorporate the discussions of many recent papers that explore the intricate complexity of the nature of semantics in the context of communications/information theory.
For example, the proposed approach based on synonymous mapping does not take into account homonymous words, which led the authors to assume no "overlap between any two synonymous sets" (p. 107).
Furthermore, the submitted manuscript does not provide any clear insight into the proposed semantic information measures, such as "semantic entropy", which seems to be basically the entropy after collecting all synonymous words together and reducing the entire "syntactic" set to the semantic information set. This is rather trivial and oversimplifies the relationship between the words/sentences and the semantics, and it also raises the question that if such a simple synonymity-based reduction were possible, why wouldn't one just apply classical Shannon theory to the semantic set? This may be what the authors are essentially doing, unfortunately it is very difficult to follow their ideas and they do not seem to have made any effort to provide insight.
The use of arithmetic coding is also a well-known approach in source coding, and there does not seem to be any remarkable novel contribution there either.
Finally, it is not clear how the authors extend their synonymity-based approach to an image communication problem, and they do not provide any numerical comparisons with any of the numerous state-of-the-art semantic image communication methods that have appeared in recent years.
Based on the above, I cannot recommend publication of this paper.
The English could be improved to more clearly express the research.
Author Response
Comment 1: It lacks convincing evidence for synonymity as a fundamental feature of semantic information (contrary to the authors' claims), as it does not sufficiently incorporate the discussions of many recent papers that explore the intricate complexity of the nature of semantics in the context of communications/information theory.
Response 1: Thank you for your question. I apologize for not realizing earlier that, given the lack of a unified perspective on semantic information, it is inappropriate to consider synonymity as a fundamental feature of semantic information. Therefore, we have avoided this statement in the revised version.
However, we would like to clarify that synonymity serves as a foundation for our understanding in semantic information, which helps establish the theoretical framework of semantic information presented by Niu et al. We state that it is an important feature for semantic information. We have added the following statement in the revised manuscript to clarify this issue:
“The basic viewpoint of synonymity aligns with Shannon's 1953 viewpoint \citep{shannon1953lattice} that ‘define the actual information of a stochastic process as that which is common to all stochastic processes which may be obtained from the original by reversible encoding operations’.” (Lines 69 ~72 in revised manuscript)
Shannon's 1953 viewpoint mentioned in our statement refers to the paper The Lattice Theory of Information. In this paper, he used sentence translation as an example to illustrate that the essence of information lies in the information (semantic information) shared by stochastic processes of different translations (syntactic information). From our perspective, different translations of a sentence represent different syntactic forms within a synonymous set, and the meaning of the sentence corresponds to the shared attribute associated with these distinct syntactic data within this synonymous set.
In addition, Shannon defined the lattice structure of information in this article to describe the different physical meanings of the abstract, sum, and product of messages. The physical meaning of the product operation, as defined in this context, represents the common information shared by different messages, which aligns closely with the actual meaning of the artificial information in the example he provided. Therefore, Shannon had considered concepts similar to synonymity early on, which aligns with our interpretation of semantic information. However, he did not fully formalize this idea into a complete theoretical system.
We acknowledge that this statement does not address all the complexities of semantic information, such as polysemy and homonymy, as you pointed out in your follow-up question. While these issues have made you question the one-to-many relationship between semantic information and syntactic information, we believe that our viewpoint based on synonymity can still provide an explanation for these phenomena. For a more detailed explanation, please refer to our response to comment 2.
In addition, regarding the issue that not discussing some recent work in semantic information theory adequately in the previous version of the manuscript, we would like to express sincerely apologize. We have now incorporated relevant consensus and discussions on recent viewpoints in the revised version (Lines 46~60) and hope these additions provide sufficient context.
Comment 2: The proposed approach based on synonymous mapping does not take into account homonymous words, which led the authors to assume no "overlap between any two synonymous sets".
Response 2: Thank you for your opinion on this issue. This is also a question that many people have raised about the theory of semantic information based on synonymity. They often question how to address the polysemy of a word, which is similar to the homonymy problem you mentioned.
To address this issue, we have added the following fundamental principle for determining synonymous mapping relationships in the revised version of the manuscript (Lines 163~165):
“It should first be considered at the sequence level, and then applied to individual elements based on contextual relationships within the sequence for practical implementation.”
This implies that when applying synonymous mapping to process semantic information, we should first place a polysemous or homonymous word within the context of a sentence to determine its specific meaning.
We present an example of a simple English sentence in the revised manuscript to support this basic principle (Lines 156~161):
“I walk along a river bank.”
In this sentence, most of the words have single meaning, while the word “bank” itself has multiple meanings, such as a financial institution and the side of a river. However, when placed in this sentence (especially placed after the word ``river''), the meaning of the word ``bank'' collapses into a specific interpretation, i.e., ``the side of the water''.
Therefore, we believe that synonymous mappings should first be processed at the sequence level before being applied at the word/element level. Besides, if a sequence cannot determine a unique meaning for each element, the sequence length must be extended until its context is sufficient to make this determination, and then treating the extended sequence as the processing object.
Actually, this principle can be confirmed based on semantic asymptotic equipartition property (AEP) proposed in \citep{niu2024Mathematical} with the assumption of sufficient long syntactic sequences. It reliably supports the proof of semantic information measures such as semantic entropy and mathematically demonstrates that when the sequence is sufficiently long, synonymous sets have no intersection or overlap.
We recognize that the length of the actual source information sequence is not always sufficient, which may occasionally prevent the determination of a specific meaning for polysemous or homonymous words/elements. Therefore, if this situation happens in real-time communication scenarios, we suggest that the communication protocol require the sender to specify a single meaning in such cases; if it happens in non-real-time scenarios, transmission can proceed in a purely syntactic manner instead of semantic information processing. While this may slightly reduce compression and communication efficiency, it ensures that the intended meaning of the original information is preserved to the greatest extent.
We hope our responses help you clarify what your concerns.
Comment 3: The submitted manuscript does not provide any clear insight into the proposed semantic information measures, such as "semantic entropy". This is rather trivial and oversimplifies the relationship between the words/sentences and the semantics, and it also raises the question that if such a simple synonymity-based reduction were possible, why wouldn't one just apply classical Shannon theory to the semantic set?
Response 3: Thank you for your comment and for providing an insight of how we define semantic information measurement and the way we process semantic information.
We acknowledge part of your opinion aligns with our understanding of semantic information processing, particularly the application of classical information theory to synonym sets. In fact, semantic information theory based on synonymy is built on this fundamental idea. However, we have found that it is not a simple matter, and some conclusions from classical information theory, such as the limits of information compression efficiency and transmission capacity, have the potential to be surpassed when processing synonymous sets. Additionally, this consideration also ensures the compatibility of semantic information processing with traditional syntactic information processing. Specifically, when it is required that there is only one element in the synonymous set (i.e., the syntactic information itself), semantic information processing will revert to classic syntactic information processing. Therefore, we do not consider this approach an oversimplification of the relationship between semantic and syntactic information.
Regarding the issue that why semantic information should be processed based on synonymous mapping by constructing synonymous sets, we have added the following statement in the revised manuscript (Lines 185–193):
“However, it should be emphasized that directly processing semantic information is not feasible, as semantic information is hidden behind the message (syntactic information) and thus invisible. Thus, semantic information processing must construct entities that can reflect semantic content. Based on the synonymous relationship we previously established, syntactic information with similar meanings can be grouped into synonymous sets, which serve as entities that reflect semantic information. Therefore, processing information at the level of these entities is then equivalent to processing semantic information. In light of this, for semantic information processing, constructing synonymous sets based on synonymous relationships is essential.”
We hope our responses help you clarify what your concerns.
Comment 4: The use of arithmetic coding is also a well-known approach in source coding, and there does not seem to be any remarkable novel contribution there either.
Response 4: Thanks for your comment. We apologize for any misunderstanding that may have been caused by the previous version of our manuscript. We would like to clarify that we are not proposing a new class of source compression algorithms. Instead, our contributions focus on developing a semantic arithmetic coding algorithm that leverages synonymity to compress semantic information, building upon the processing of classical arithmetic coding. Therefore, the novel contribution does not lie in the logic of arithmetic coding itself but in the design of the algorithm that applies arithmetic coding to process semantic information.
Comment 5: It is not clear how the authors extend their synonymity-based approach to an image communication problem, and they do not provide any numerical comparisons with any of the numerous state-of-the-art semantic image communication methods that have appeared in recent years.
Response 5: Thank you for your suggestion. However, we apologize if our previous explanation in the Introduction section of our previous manuscript may have inadvertently misled you regarding the focus of our work. We would like to emphasize that the core of our work is to address a semantic information compression problem rather than an end-to-end semantic communication problem. Therefore, we reorganized the narrative structure of the Introduction section and extensively rewrote it to better emphasize the focus of our work.
We would like to clarify that since our work focuses solely on semantic source compression, it would be inappropriate to compare our approach with semantic communication methods. It is foreseeable that the presence of arithmetic coding will inevitably result in a cliff effect in the low SNR range, which is a common issue in separate source-channel coding schemes and not exists in recent deep JSCC-based semantic communication methods. Therefore, a performance comparison with semantic communication methods is beyond the scope of our study.
However, as you mentioned, this work has the potential to be extended into a semantic communication method, which is also a key research direction for our future work. In our initial conception, to avoid the cliff effect, this extension cannot rely on arithmetic codes. However, the semantic information processing logic that approximates semantic entropy can be adapted to the design of semantic communication methods. In the revised version of the manuscript, we have included this under future works in the Conclusion section.
We hope our responses help you clarify what your concerns.
Reviewer 2 Report
Comments and Suggestions for Authors
This paper introduces a new epistemology for semantic information theory, proposing that synonymity is the fundamental feature of semantics and using synonymous mappings to connect semantic and syntactic information. Based on this, the authors present a Semantic Arithmetic Coding (SAC) method that constructs synonymous mappings to partition sets and performs arithmetic coding over these sets, achieving higher compression efficiency than traditional methods. Experiments on edge texture map compression demonstrate significant improvements in coding efficiency without semantic losses, confirming the practical effectiveness of SAC. In general, I think this paper is well written with solid contribution. The quality of this paper can be further improved if the following comments can be addressed.
First, as shown in Fig.1, this paper assumes that semantic elements can be mapped to an equal number of synonymous sets, representing different meaning and without overlap. This assumption needs to be further justified. What if they are mapped to the overlapped syntactic set? How to adjust the proposed SAC to the general case?
The definition of semantically lossless needs further elaboration. It seems that the semantical lossless is only possible if the mapping from the semantic information set to lower entropy syntactic information set. This is not attributed to the proposed coding strategy, but to the communication purpose itself, whose validity depends on the scenarios.
It is interesting to discuss the insights on the mapping from semantic information to syntactic information. How can we perform this kind of mapping in a general scenario? Is it guaranteed that the entropy will reduce after mapping?
The complexity of the encoding and decoding algorithm for the proposed SAC should be discussed.
Author Response
Comment 1: As shown in Fig.1, this paper assumes that semantic elements can be mapped to an equal number of synonymous sets, representing different meaning and without overlap. This assumption needs to be further justified. What if they are mapped to the overlapped syntactic set? How to adjust the proposed SAC to the general case?
Response 1: Thanks for your comment, which is very helpful to improve our work. In the previous version of our manuscript, we used Figure 1 to show that each element can use synonymous mapping to construct synonymous set with different meanings and without overlap. This might raise questions, such as how to address phenomena like polysemy or homonymy, which will cause the overlapping between these sets. These are common concerns regarding the basic theory we are considering.
However, we would like to clarify that in the previous version, we overlooked a basic criterion for determining synonymous mapping relationships, i.e., It should first be considered at the sequence level, and then applied to individual elements based on contextual relationships within the sequence for practical implementation. In our revised version, we supplement this basic principle (Lines 162-173) and illustrate it with a simple example of English sentence (Lines 155-161). Based on this example, we can see that the contextual structure or Markov chain characteristics of the sequence help determine a unique meaning for polysemous or homonymous words/elements, which will prevent overlap between synonymous sets.
Since semantic arithmetic coding operates on sequences, as long as the information sequence does not satisfy the i.i.d. property, the operation is in the general case rather than the simplified one, so that the SAC encoding can directly solve the above problem. This is because, during the operation of semantic arithmetic coding, the algorithm determines the synonymous mapping relationship symbol by symbol which based on the context structure or Markov chain characteristics, and calculates the probability for the synonymous sets. Therefore, the SAC algorithm we proposed can handle the general case effectively.
Comment 2: The definition of semantically lossless needs further elaboration. It seems that the semantical lossless is only possible if the mapping from the semantic information set to lower entropy syntactic information set. This is not attributed to the proposed coding strategy, but to the communication purpose itself, whose validity depends on the scenarios.
Response 2: Thank you for providing your insights on this problem. Your insights align with our understanding to some extent, especially using the lower entropy syntactic information set to guarantee the semantic lossless. However, we want to clarify that we do not map the semantic information set to a lower entropy syntactic set: We map the higher entropy original syntactic set to the lower entropy syntactic set, in which the latter one is corresponding to the semantic information that the original syntactic sequences/variables express. In conjunction with the response 1, we would like to emphasize that this method is independent of the encoding strategy. Since the basic principle of determining synonymous mapping relationships is priority to the sequence level, as long as there is semantic information within the syntactic sequence, we are inevitably able to construct synonymous sets for this syntactic sequence, and thus reducing the entropy while only required to maintaining the semantic information.
In addition, we would like to thank you for revealing a basic logic of our semantic information processing: Semantic information is invisible, and processing semantic information necessarily requires processing the syntactic information entities corresponding to the semantic information (i.e., mapping to the lower entropy syntactic set that you mentioned). In light of this, we supplement the corresponding expressions in our revised manuscript (Lines 184~193):
“However, it should be emphasized that directly processing semantic information is not feasible, as semantic information is hidden behind the message (syntactic information) and thus invisible. Thus, semantic information processing must construct entities that can reflect semantic content. Based on the synonymous relationship we previously established, syntactic information with similar meanings can be grouped into synonymous sets, which serve as entities that reflect semantic information. Therefore, processing information at the level of these entities is then equivalent to processing semantic information. In light of this, for semantic information processing, constructing synonymous sets based on synonymous relationships is essential.”
We hope our responses help you clarify what your concerns.
Comment 3: It is interesting to discuss the insights on the mapping from semantic information to syntactic information. How can we perform this kind of mapping in a general scenario? Is it guaranteed that the entropy will reduce after mapping?
Response 3: Thank you for acknowledging our viewpoint. However, I am not sure whether the general case you mentioned refers to source information in a broader sense, such as text sequences, images, audio, etc.? If our understanding is correct, we would like to state a possible solution:
Since synonymous mappings determine common semantic information for all the syntactic data in the synonymous set, a very likely situation is that these syntactic data share similar or identical implicit representations. Therefore, this method of constructing synonymous sets corresponding to the synonymous mappings may be realized through neural network training. We are currently conducting research in this direction.
As for whether entropy will decrease, according to our response 1 and the form we define semantic entropy, the semantic entropy obtained after processing will be definitely lower than the classical Shannon entropy of the original syntactic information, which will inevitably improve the compression efficiency while maintaining the semantic information.
Comment 4: The complexity of the encoding and decoding algorithm for the proposed SAC should be discussed.
Response 4: Thanks for your suggestions. In our revised manuscript, we supplied detailed computational complexity of SAC encoding and decoding algorithm in Section 3.1 (Lines 261~279) and Section 3.2 (Lines 320~346). We conclude the comparison of the computational complexities between our proposed SAC algorithm and the general arithmetic coding (Lines 347~350), i.e.,
“The computational complexity of SAC encoding is generally higher than that of general arithmetic encoding but be lower than that in the i.i.d. case, while the computational complexity of SAC decoding is lower than that of general arithmetic decoding in all cases.”
We thank you again for your suggestions, which helped improve our work. We hope our revisions clarify your confusions.
Round 2
Reviewer 1 Report
Comments and Suggestions for Authors
The authors have addressed my concerns. I think the paper is now in good shape.